# Token-Regulated Group Relative Policy Optimization for Stable Reinforcement Learning in Large Language Models

## Abstract

Reinforcement learning with verifiable rewards (RLVR) has emerged as a powerful approach for strengthening the reasoning capabilities of large language models (LLMs). Among existing algorithms, Group Relative Policy Optimization (GRPO) has demonstrated strong performance, yet it suffers from a critical issue: low-probability tokens disproportionately dominate gradient updates due to their inherently large gradient magnitudes. This imbalance leads to unstable training and suppresses the contribution of high-probability tokens that are more reliable for learning. In this work, we introduce **Token-Regulated Group Relative Policy Optimization (TR-GRPO)**, a simple yet effective extension of GRPO that assigns token-level weights positively correlated with the model's predicted probability. By downweighting low-probability tokens and emphasizing high-probability ones, TR-GRPO mitigates gradient over-amplification while preserving informative learning signals. We provide theoretical analysis to show how token-level probability governs gradient norms which motivates our weighting design. Extensive experiments demonstrate that TR-GRPO consistently outperforms GRPO across RLVR tasks—including logic, math, and agentic reasoning—highlighting the importance of regulating token contributions during RL training and establishing TR-GRPO as a robust framework for enhancing LLM reasoning.

## 1 Introduction

Large language models (LLMs) have recently demonstrated remarkable progress in complex reasoning tasks such as mathematics and programming, with systems like OpenAI's O-series (Jaech et al., 2024), DeepSeek-R1 (Guo et al., 2025), Kimi K2 (Team et al., 2025), and Qwen3 (Yang et al., 2025a) achieving state-of-the-art performance. A central driver of these advances is Reinforcement Learning with Verifiable Rewards (RLVR) (Gao et al., 2024; Lambert et al., 2024; Team et al., 2025; Guo et al., 2025; Yang et al., 2025a), which commonly employs reinforcement learning on an LLM with a rule-based outcome reward, such as a binary indicator reward for mathematical or logical validity. RLVR has emerged as a practical framework for strengthening the reasoning abilities of LLMs, offering stable and task-aligned reward signals without the need for costly human annotation or additional models. Among the RL algorithms used in this setting, Group Relative Policy Optimization (GRPO) (Shao et al., 2024) has become a popular choice due to its simplicity and strong empirical results.

Although recent progress in RLVR has been driven by new algorithms (Yu et al., 2025; Yue et al., 2025; Guan et al., 2025), cross-domain applications (Xue et al., 2025; Liu et al., 2025; Pan et al., 2025), and unexpected empirical observations (Wang et al., 2025; Yue et al., 2025; Zhao et al., 2025), current implementations typically still treat all tokens equally without considering which ones are truly important, overlooking the fact that model responses often contain a mixture of both high-quality and low-quality tokens, especially as response length increases. Such uniform training disregards the varied functional roles tokens play in reasoning, which can hinder performance by neglecting critical tokens and amplifying noise from less relevant ones.

(a) Frequent tokens with the highest average probability.

(b) Frequent tokens with the lowest average probability.

Figure 1: Word clouds of the top 100 high- vs. low-probability tokens selected from frequently occurring words. High-probability tokens (left) primarily consist of mathematical and logical operators, brackets, and variable names, where even small errors can invalidate an entire solution, whereas low-probability tokens (right) mostly consist of generic content words that are less critical.

Motivated by these observations, we ask: *which tokens actually drive the update during RLVR?*. Our analysis (Theorem 3.1) shows that the gradient norm of a token scales with $1 - \pi_\theta(o_t)$, where $o_t$ is the token and $\pi_\theta(o_t)$ is its model probability; hence, low-probability tokens induce larger gradients and more aggressive updates, while high-probability tokens induce smaller, more conservative updates. To check whether this aligns with semantic importance, we visualize the top 100 frequent high probability tokens and low-probability tokens ranked by their average model probability $\bar{\pi}(o_t) = \frac{1}{\#\text{occ}(o_t)} \sum_{o_t} \pi_\theta(o_t)$ (Figures 1a–1b) where $\#\text{occ}(o_t)$ denotes the number of occurrences of token $o_t$. Figure 1a shows that high-probability tokens concentrate on mathematical and logical *structure*—operators, brackets, variable names, and formatting markers—where even a single error can break the entire solution, whereas low-probability tokens are mostly generic content words (e.g., "output," "particular," "location") that contribute less to the core reasoning and are more easily replaced without changing meaning. This creates a clear tension: standard GRPO naturally magnifies low-probability tokens (via larger gradients) even though high-probability tokens carry the most critical signal for correctness. We therefore need a mechanism to *mitigate* over-updates on low-probability tokens while preserving and even *reinforcing* updates on high-probability ones.

To tackle this, we propose a simple yet effective method: **Token-Regulated GRPO**, which introduces a probability-aware token weight that increases with the token's probability as described in Eq. (9), so that high-probability tokens are emphasized and low-probability ones are downweighted. Experiments across logic puzzles, mathematical reasoning, and agentic QA confirm that Token-Regulated GRPO consistently outperforms GRPO. In particular, across all these RLVR settings, TR-GRPO delivers substantial gains: on the K&K logic puzzles benchmark, Qwen2.5-3B improves from **0.39** to **0.63** (Table 1); in agentic multi-hop QA, exact-match accuracy nearly doubles from **13.84** to **27.29** (Table 3); and on math benchmarks including OlympiadBench and Minerva, we observe average improvements of **+6–10%** (Table 2). Beyond raw accuracy, TR-GRPO yields smoother gradient trajectories and more stable training dynamics (Figure 3), confirming the benefit of probability-aware reweighting.

## 2 BACKGROUND

### 2.1 LARGE LANGUAGE MODELS

Most **Large Language Models (LLMs)** adopt a transformer decoder-only architecture (Vaswani et al., 2017), parameterized by $\theta \in \mathbb{R}^d$ and denoted as $\pi_\theta$. The basic unit of an LLM is the token, which may correspond to a word, subword, or character, and is selected from a finite vocabulary $\mathcal{V} = \{v^1, \dots, v^N\}$ of size $N$. Given a prompt $q$, the model generates a sequence of tokens $o = (o_1, \dots, o_T)$ autoregressively. At each step $t$, the model produces a distribution over $\mathcal{V}$ conditioned on $q$ and the previously generated tokens $o_{<t}$, from which the next token is sampled:

$$o_t \sim \pi_\theta(\cdot \mid q, o_{<t}). \tag{1}$$

This process continues until an end-of-sequence (EOS) token is produced or the sequence length reaches a predefined maximum $t_{\max}$.

While pretrained and supervised fine-tuned LLMs achieve fluency, practical applications often require alignment with human preferences or reasoning ability, which is not easily captured by likelihood training alone. To address this, one may introduce a reward function $r(q, o)$ that quantifies the quality of a generated sequence $o$ given prompt $q$. The learning problem can then be cast as reinforcement learning, where the prompt $q$ represents the state, each generated token is an action, and the full output $o$ yields a terminal reward. Accordingly, the optimization objective is formulated as: $\max_\theta \mathbb{E}_{q \sim \mathcal{Q}, \, o \sim \pi_\theta(\cdot|q)} \left[ r(q, o) \right]$, where $\mathcal{Q}$ denotes a dataset of prompts.

## 2.2 REINFORCEMENT LEARNING WITH VERIFIABLE REWARDS (RLVR)

**Reinforcement Learning with Verifiable Rewards (RLVR)** (Gao et al., 2024; Lambert et al., 2024; Team et al., 2025; Guo et al., 2025; Yang et al., 2025a) provides a framework for optimizing LLMs in domains where correctness can be *deterministically verified*. Typical applications include code generation validated by unit tests, mathematical problem solving with symbolic checkers, and factual QA with exact-match rules or programmatic validators. Unlike preference-based reinforcement learning, RLVR leverages a predefined verifier to produce a reward without requiring human labels. Formally, given a dataset of prompts $\mathcal{D}$, a policy $\pi_\theta$, and a frozen reference model $\pi_{\text{ref}}$, the objective is:

$$\max_\theta \mathbb{E}_{q \sim \mathcal{Q}, \, o \sim \pi_\theta(\cdot|q)} \left[ R_\phi(q, o) \right] - \beta \, \mathbb{D}_{\text{KL}} [\pi_\theta(o \mid q) \, \| \, \pi_{\text{ref}}(o \mid q)], \quad (2)$$

where $R_\phi$ is the verifiable reward function and $\beta$ controls the KL regularization. A typical verifiable reward function is defined as:

$$R_\phi(q, o) = \begin{cases} 1 & \text{if } \text{match}(o, o_g), \\ -1 & \text{otherwise}, \end{cases} \quad (3)$$

where $o_g$ is the ground-truth answer and $\text{match}(\cdot, \cdot) \in \{0, 1\}$ indicates whether the generated output matches. More generally, $R_\phi$ may be graded (e.g., partial credit, length penalty, latency) while remaining *verifiable* by rule-based approaches (Wang et al., 2024) or model-based verifiers (Ma et al., 2025).

In summary, RLVR provides a principled setting where optimization is guided by *verifiable correctness*, making it a natural foundation for training reasoning-oriented LLMs.

## 2.3 GROUP RELATIVE POLICY OPTIMIZATION (GRPO)

**Group Relative Policy Optimization (GRPO)**, first introduced by Shao et al. (2024), is a policy-gradient algorithm widely used for optimizing the objective in Eq. 2. Unlike the popular PPO algorithm (Schulman et al., 2017), which requires training an additional value function alongside the LLM, GRPO removes the dependency on an explicit value model. Instead, it estimates advantages by *normalizing rewards within a group* of sampled responses to the same prompt. Specifically, for a prompt $q$ with $G$ sampled responses $\{o_i\}_{i=1}^G$ and associated scalar rewards $\{r_i\}_{i=1}^G$, GRPO defines a group-normalized advantage as:

$$\hat{A}_{i,t} = \frac{r_i - \text{mean}(\{r_j\}_{j=1}^G)}{\text{std}(\{r_j\}_{j=1}^G)}, \quad (4)$$

The effectiveness of the above normalization method can be interpreted through the lens of reward shaping: by emphasizing relative differences among candidate outputs for the same prompt, it strengthens the stability of the gradient signal and maintains its reliability, even under sparse reward conditions (Hu et al., 2020). In this study, we employ a variant of GRPO to optimize the policy model $\pi_\theta$. The optimization objective of GRPO can be expressed as follows:

$$\mathcal{J}_{\text{GRPO}}(\theta) = \mathbb{E}\big[q \sim \mathcal{Q}, \, \{o_i\}_{i=1}^G \sim \pi_{\theta_{\text{old}}}(O \mid q)\big]$$

$$\frac{1}{\sum_{i=1}^G |o_i|} \sum_{i=1}^G \sum_{t=1}^{|o_i|} \Big\{ \min\big(r_{i,t}(\theta)\, \hat{A}_{i,t}, \, \text{clip}(r_{i,t}(\theta), \, 1 - \epsilon_l, \, 1 + \epsilon_h)\, \hat{A}_{i,t}\big) \, - \, \beta\, \mathbb{D}_{\text{KL}}[\pi_\theta \, \| \, \pi_{\text{ref}}] \Big\}, \quad (5)$$

where $r_{i,t}(\theta) = \frac{\pi_\theta(o_{i,t}|q, o_{i,<t})}{\pi_{\theta_{\text{old}}}(o_{i,t}|q, o_{i,<t})}$ is the importance sampling ratio and the KL divergence term is defined as:

$$\mathbb{D}_{\text{KL}}[\pi_\theta \, \| \, \pi_{\text{ref}}] = \frac{\pi_{\text{ref}}(o_{i,t} \mid q, o_{i,<t})}{\pi_\theta(o_{i,t} \mid q, o_{i,<t})} - \log \frac{\pi_{\text{ref}}(o_{i,t} \mid q, o_{i,<t})}{\pi_\theta(o_{i,t} \mid q, o_{i,<t})} - 1. \quad (6)$$

Here, $\pi_{\theta_{\text{old}}}$ denotes the policy used to sample responses, $\pi_{\text{ref}}$ is a frozen reference model, and $\epsilon_l, \epsilon_h$ are the PPO-style clipping threshold hyper-parameters and $\beta$ controls KL regularization.

## 3 METHODOLOGY

We now present our proposed approach. To see what kinds of tokens highly impact the update, we start with the derivative of the GRPO's objective function w.r.t. the model parameters.

$$\nabla_\theta J_{\text{GRPO}}(\theta) = \mathbb{E}_{q \sim \mathcal{D}, \{o_i\}_{i=1}^G}$$

$$\left[ \frac{1}{\sum_{i=1}^G |o_i|} \sum_{i=1}^G \sum_{t=1}^{|o_i|} \underbrace{\left( \frac{\pi_\theta(o_{i,t})}{\pi_{\text{old}}(o_{i,t})} \hat{A}_{i,t} \cdot \mathbb{I}_{\text{trust}}\left( \frac{\pi_\theta(o_{i,t})}{\pi_{\text{old}}(o_{i,t})}, \hat{A}_{i,t} \right) + \beta \frac{\pi_{\text{ref}}(o_{i,t})}{\pi_\theta(o_{i,t})} - \beta \right)}_{\gamma_{i,t}} \nabla_\theta \log \pi_\theta(o_{i,t}) \right]$$

$$(7)$$

where for simplicity, we denote $\pi_\theta(o_{i,t}) := \pi_\theta(o_{i,t} \mid q, o_{i,<t})$ and define

$$\mathbb{I}_{\text{trust}}\left( \frac{\pi_\theta(o_{i,t})}{\pi_{\text{old}}(o_{i,t})}, \hat{A}_{i,t} \right) = \begin{cases} 0, & \text{if } \hat{A}_{i,t} > 0 \text{ and } \frac{\pi_\theta(o_{i,t})}{\pi_{\text{old}}(o_{i,t})} > 1 + \epsilon_h, \\ 0, & \text{if } \hat{A}_{i,t} < 0 \text{ and } \frac{\pi_\theta(o_{i,t})}{\pi_{\text{old}}(o_{i,t})} < 1 - \epsilon_l, \\ 1, & \text{otherwise.} \end{cases}$$

We represent our LLM as the composition of many layers $f = f_L \circ f_{L-1} \circ ... \circ f_l \circ ... \circ f_1$ where $f$ maps from the input sequence of tokens $a_0$ to the output sequence of tokens $a_L$. Moreover, the output sequence of tokens $a_L$ is used to compute the logit $h_{i,t}$ for predicting the token $o_{i,t}$. Let us denote $a_{l-1}$ and $a_l$ as the input and output of the $l$-th layer, i.e., $a_l = f_l(a_{l-1}, \theta_l)$ where $\theta_l$ represents the model parameter at the $l$-th layer. To quantify the gradient for update, we develop the following theorem.

**Theorem 3.1.** *Let us denote the classifier head at the output layer, the Jacobian* $\frac{\partial f_l(a_{l-1}, \theta_l)}{\partial a_{l-1}}$, *and the gradient matrix* $\frac{\partial f_l(a_{l-1}, \theta_l)}{\partial \theta_l}$ *as* $W$, $J_l$, *and* $G_l$ *respectively. Moreover, we further assume that* $\sigma_{\text{min}}(W) \geq (\alpha^W)^2$, $\sigma_{\text{max}}(W) \leq (\beta^W)^2$, $\sigma_{\text{min}}(J_l) \geq (\alpha_l^J)^2$, $\sigma_{\text{max}}(J_l) \leq (\beta_l^J)^2$, *and* $\sigma_{\text{min}}(G_l) \geq (\alpha_l^G)^2$, $\sigma_{\text{max}}(G_l) \leq (\beta_l^G)^2$ *where all lower/upper bounds are positive and* $\sigma_{\text{min}}(\cdot)$, $\sigma_{\text{max}}(\cdot)$ *return the smallest and the largest singular values of a matrix. We can bound the gradient norm* $\|\nabla_\theta \log \pi_\theta(o_{i,t})\|_2$ *as*

$$\frac{(1 - \pi_\theta(o_{i,t}))}{\sqrt{L}} \sum_{l=1}^L \left( a^W a_l^G \prod_{i=l}^L a_i^J \right) \leq \|\nabla_\theta \log \pi_\theta(o_{i,t})\|_2 \leq \sqrt{2}(1 - \pi_\theta(o_{i,t})) \sum_{l=1}^L \left[ b^W b_l^G \prod_{i=l}^L b_i^J \right].$$

$$(8)$$

From the bounds in (8), it appears that low-probability tokens with a small $\pi_\theta(o_{i,t})$ tend to have higher gradient norms $\|\nabla_\theta \log \pi_\theta(o_{i,t})\|_2$, leading to more aggressive updates in Eq. (7), while high-probability tokens with a high $\pi_\theta(o_{i,t})$ tend to have smaller gradient norms $\|\nabla_\theta \log \pi_\theta(o_{i,t})\|_2$, leading to more humble updates.

According to the above observation, we should govern the update of low/high-probability tokens in the way that reduces the aggressiveness of low-probability token updates and maintains the consistent updates for high-probability tokens. Hinting at this motivation, we propose our approach **Token-Regulated GRPO (TR-GRPO)**.

$$\mathcal{J}_{\text{TR-GRPO}}(\theta) = \mathbb{E}\big[q \sim \mathcal{Q}, \ \{o_i\}_{i=1}^G \sim \pi_{\theta_{\text{old}}}(O \mid q)\big]$$

$$\frac{1}{\sum_{i=1}^G |o_i|} \sum_{i=1}^G \sum_{t=1}^{|o_i|} \Big\{ \min\big(w_{i,t} r_{i,t}(\theta) \hat{A}_{i,t}, \ \text{clip}(w_{i,t} r_{i,t}(\theta), 1 - \epsilon_l, 1 + \epsilon_h) \hat{A}_{i,t}\big) \quad (9)$$

$$- \beta \, \mathbb{D}_{\text{KL}}\big[w_{i,t}; \pi_\theta \| \pi_{\text{ref}}\big] \Big\},$$

where we have defined

$$\mathbb{D}_{\mathrm{KL}}[w_{i,t}; \pi_\theta \,\|\, \pi_{\mathrm{ref}}] = w_{i,t} \frac{\pi_{\mathrm{ref}}(o_{i,t} \mid q, o_{i,<t})}{\pi_\theta(o_{i,t} \mid q, o_{i,<t})} - \log(w_{i,t} \frac{\pi_{\mathrm{ref}}(o_{i,t} \mid q, o_{i,<t})}{\pi_\theta(o_{i,t} \mid q, o_{i,<t})}) - 1. \quad (10)$$

$$w_{i,t} = \mathrm{clip}\big(\alpha \cdot \big[\sigma\big(\frac{\mathrm{sg}[\pi_\theta(o_{i,t} \mid q, o_{i,<t})]}{\tau}\big) - \mu\big], \, L, \, U\big), \quad (11)$$

in which $\mathrm{sg}[\cdot]$ denotes only taking the numerical value but stopping the gradient, corresponding to the detach operation in PyTorch, $\alpha$, $\mu$ and $\tau$ are constant, while $L$ and $U$ are lower and upper clipping bounds.

To determine which tokens should be prioritized during the policy update, we compare two scenarios: (i) **TR-GRPO**, which assigns higher weights to high-probability tokens and lower weights to low-probability ones; and (ii) a **Reverse** variant that inverts TR-GRPO's estimate by assigning $2 - w$ to any token whose TR-GRPO weight is $w$, thereby giving lower weights to high-probability tokens and higher weights to low-probability ones. As shown in Figure 2, **TR-GRPO (blue)** exhibits steadier accuracy gains and consistently outperforms both **Reverse (orange)** and the **GRPO baseline (green)** across K&K puzzle sizes (3–7), whereas Reverse tracks GRPO closely without clear improvement. These results reinforce our motivation that emphasizing high-probability tokens leads to more stable and effective learning than over-amplifying low-probability ones.

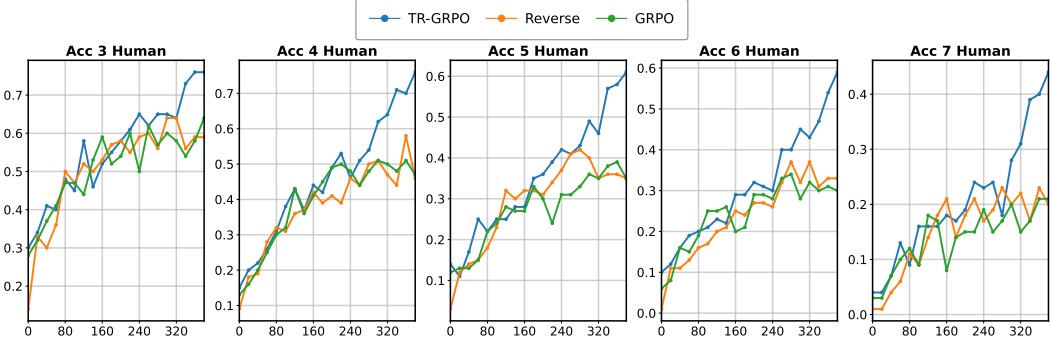

Figure 2: Accuracy on the K&K Logic Puzzles benchmark, broken down by puzzle size (3–7 people). TR-GRPO consistently achieves higher accuracy than GRPO across all difficulty levels, while the Reverse variant that emphasizes low-probability tokens yields performance comparable to GRPO without clear improvement.

To realize what low/high-probability tokens are, in Figures 1, we visualize the top 100 low/high-probability tokens in each category. Figure 1a shows that high-probability tokens are often critical mathematical or logical symbols and formatting markers (e.g., operators, brackets, variable names). These elements are semantically indispensable: even a single error can invalidate an entire solution, meaning they are hardly replaceable. In contrast, Figure 1b illustrates that low-probability tokens are mostly generic, content words such as "output," "particular," or "location," which contribute less to the core logical meaning of a sentence and can usually be substituted without altering the semantics. This contrast again provides intuition for why emphasizing high-probability tokens is more sensible—prioritizing them ensures stable updates on the tokens that carry the most critical signal.

We now investigate the derivative of the objective function in our approach, which has the following form

$$\nabla_\theta J_{\mathrm{TR\text{-}GRPO}}(\theta) = \mathbb{E}_{q \sim \mathcal{D}, \{o_i\}_{i=1}^G} \quad (12)$$

$$\left[ \frac{1}{\sum_{i=1}^G |o_i|} \sum_{i=1}^G \sum_{t=1}^{|o_i|} \underbrace{\left( \frac{\pi_\theta(o_{i,t})}{\pi_{\mathrm{old}}(o_{i,t})} \hat{A}_{i,t} \cdot \mathbb{I}_{\mathrm{trust}}\left( \frac{\pi_\theta(o_{i,t})}{\pi_{\mathrm{old}}(o_{i,t})}, \hat{A}_{i,t} \right) + \beta \frac{\pi_{\mathrm{ref}}(o_{i,t})}{\pi_\theta(o_{i,t})} - \beta \right)}_{\gamma_{i,t}} w_{i,t} \nabla_\theta \log \pi_\theta(o_{i,t}) \right],$$

where we have defined

$$
\mathbb{I}_{\text{trust}}\left(\frac{\pi_\theta(o_{i,t})}{\pi_{\text{old}}(o_{i,t})}, \hat{A}_{i,t}\right) = \begin{cases} 0, & \text{if } \hat{A}_{i,t} > 0 \text{ and } \frac{\pi_\theta(o_{i,t})}{\pi_{\text{old}}(o_{i,t})} > w_{i,t}^{-1}\left(1 + \epsilon_h\right), \\ 0, & \text{if } \hat{A}_{i,t} < 0 \text{ and } \frac{\pi_\theta(o_{i,t})}{\pi_{\text{old}}(o_{i,t})} < w_{i,t}^{-1}\left(1 - \epsilon_l\right), \\ 1, & \text{otherwise.} \end{cases}
$$

Denote $g_{i,t} = \gamma_{i,t} w_{it} \nabla_\theta \log \pi_\theta(o_{i,t})$ as the token-based derivative in Eq. (12) relevant to $o_{i,t}$, we have the following theorem.

**Theorem 3.2.** *Under the same assumptions in Theorem 3.1, we can bound $\|g_{i,t}\|_2$ as*

$$
\frac{w_{i,t}\left(1 - \pi_\theta\left(o_{i,t}\right)\right)}{\sqrt{L}}\left|\gamma_{i,t}\right|\sum_{l=1}^{L}\left(a^W a_l^G \prod_{i=l}^{L} a_i^J\right) \leq \|g_{i,t}\|_2 \leq \sqrt{2} w_{i,t}\left(1 - \pi_\theta\left(o_{i,t}\right)\right)\left|\gamma_{i,t}\right|\sum_{l=1}^{L}\left[b^W b_l^G \prod_{i=l}^{L} b_i^J\right].
$$

$$(13)$$

According to the bounds in (13), for low-probability tokens, $1 - \pi_\theta\left(o_{i,t}\right)$ is high and $w_{i,t}$ is low, whereas for high-probability tokens, $1 - \pi_\theta\left(o_{i,t}\right)$ is low and $w_{i,t}$ is high. As a result, the product $w_{i,t}\left(1 - \pi_\theta\left(o_{i,t}\right)\right)$ becomes more stable (i.e., exhibits lower variance) for both low-probability and high-probability tokens. Therefore, in our approach, $\|g_{i,t}\|_2$ may have lower variance for both low-probability and high-probability tokens, leading to a lower variance in the gradient norm compared to GRPO. To demonstrate this theoretical finding, we plot the gradient norms during training under three RLVR settings (described in detail in Section 4) in Figure 3. Compared to GRPO, TR-GRPO exhibits noticeably smaller fluctuations and fewer outlier spikes, resulting in a smoother and more stable optimization trajectory that aligns with our theoretical analysis.

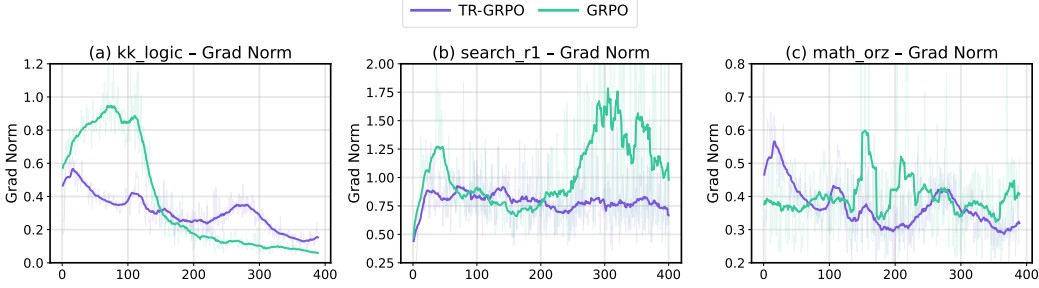

Figure 3: Gradient norm trajectories during training under GRPO vs. TR-GRPO across three RLVR settings. TR-GRPO consistently exhibits lower variability and fewer spikes than GRPO, indicating stabilized updates in accordance with the bound in Eq. (13).

In addition to gradient norm analysis, we also track training rewards on Qwen2.5-3B-Instruct across logic, math, and agentic settings. As shown in Figure 4, TR-GRPO consistently yields higher and more stable reward trajectories than GRPO, further supporting the advantages of our proposed method in maintaining and encouraging the generation of high-probability tokens to form high-rewarded responses.

## 4 EXPERIMENTS

We conduct extensive experiments across multiple RLVR benchmarks to assess the effectiveness of TR-GRPO. The results demonstrate that our method consistently outperforms GRPO, delivering stronger reasoning ability and more stable training dynamics.

### 4.1 EXPERIMENTAL SETUP

To validate the effectiveness and generality of our proposed method, we conduct experiments in three widely used RLVR settings that stress different aspects of reasoning: (i) Logic, (ii) Math, and (iii) Agentic.

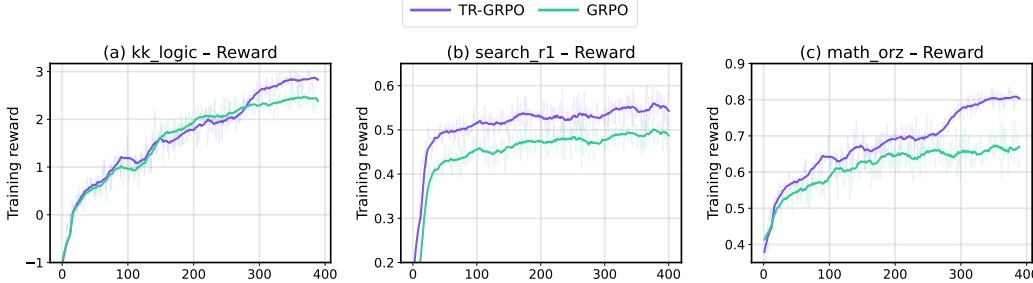

Figure 4: Training Reward trajectories during training under GRPO vs. TR-GRPO across three RLVR settings. The figure shows that TR-GRPO consistently achieves higher reward throughout training, indicating more effective learning and supporting our claim that probability-aware token weighting improves reasoning ability.

For the **logic** data, we adopt the K&K (Knights and Knaves) logic puzzles introduced in (Xie et al., 2024). Each puzzle describes a set of people who always lie or always tell the truth, and the task is to determine their identities given a set of statements. For training, we use the same dataset splits as in (Yang et al., 2025c), which include synthetic instances of increasing difficulty from 3 to 7 (measured by the number of people per puzzle). The reward function checks both the output format (requiring explicit `<think>` and `<answer>` tags) and the correctness of the final assignment. During evaluation, we follow the official harness and report accuracy across all difficulty levels, as well as breakdowns by puzzle size. This setting emphasizes step-by-step deductive reasoning and tests whether the model can align reasoning traces with verifiable logical consistency.

We next consider **mathematical** reasoning tasks, where correctness can be verified automatically against gold-standard answers. Following (Yang et al., 2025c), we train on datasets that combine symbolic manipulation and arithmetic word problems, using a binary reward signal: 1 if the final boxed answer matches the reference solution and 0 otherwise. The training corpus includes diverse math reasoning prompts designed to encourage structured derivations rather than direct guessing. For evaluation, we adopt multiple benchmarks that are standard in math-focused RLVR research: `Olympiad Bench` Hendrycks et al. (2021), `Minerva` (Lewkowycz et al., 2022), `MATH-500` (He et al., 2024), `AMC 2022-2023` and `AIME 2024`. For the first three benchmarks, evaluation is conducted using greedy decoding. For the last two benchmarks, consistent with standard practice, we generate 16 responses per question and report the mean accuracy across these samples (avg@16). Notably, since `AIME 2024` is an extremely challenging dataset, we also report pass@16, which counts a problem as solved if at least one of the 16 responses is correct.

For **agentic** reasoning tasks, we examine an agentic setting that requires knowledge-intensive reasoning augmented with retrieval tools. Following the **Search-R1** (Jin et al., 2025) configuration from **VerlTool** framework (Jiang et al., 2025), the model interacts with an external tool server providing a retriever and other utilities such as Python or SQL. Training data consists of open-domain QA datasets where each instance requires grounding reasoning steps with retrieved evidence. The reward is based on exact-match correctness of the final answer, while intermediate tool calls are masked in the policy loss to avoid leakage of supervision. For evaluation, we follow the VerlTool benchmark and report exact-match accuracy on both General Q&A benchmarks (`NQ` (Kwiatkowski et al., 2019), `TriviaQA` (Joshi et al., 2017), `PopQA` (Mallen et al., 2022)) and multi-hop Q&A benchmark (`HotpotQA` (Yang et al., 2018), `2Wiki` (Ho et al., 2020), `MuSiQue` (Trivedi et al., 2022), `Bamboogle` (Press et al., 2022))).

We leave further implementation details and hyperparameters to Appendix B with detailed evaluation setting adopted by our experiments.

## 4.2 MAIN RESULTS

Besides the detailed analysis of low and high-probability tokens in Section 3, we now present the main empirical results of our study. Across all three widely used RLVR settings include logic, math, and agentic, our proposed TR-GRPO method achieves substantial and consistent gains over the widely used GRPO baseline.

Table 1: Experimental results on the K&K Logic Puzzles benchmark. The best results are indicated in **bold**.

| Model | Difficulty by Number of People | | | | | Avg. | |
| | 3 | 4 | 5 | 6 | 7 | | |
|---|---|---|---|---|---|---|---|
| GPT-4o | 0.57 | 0.49 | 0.32 | 0.23 | 0.21 | 0.36 | |
| o1-2024-12-17 | 0.51 | 0.38 | 0.38 | 0.35 | 0.30 | 0.38 | |
| Deepseek-R1 | 0.73 | 0.77 | 0.78 | 0.75 | 0.88 | 0.78 | |
| Qwen2.5-3B-Instruct | 0.09 | 0.10 | 0.03 | 0.05 | 0.02 | 0.06 | |
| + GRPO | 0.64 | 0.47 | 0.35 | 0.30 | 0.21 | 0.39 | |
| + TR-GRPO | **0.76** | **0.76** | **0.61** | **0.59** | **0.44** | **0.63** | ↑ 61.5% |
| Qwen2.5-7B-Instruct-1M | 0.22 | 0.15 | 0.08 | 0.10 | 0.02 | 0.11 | |
| + GRPO | 0.91 | 0.91 | 0.77 | 0.65 | 0.61 | 0.77 | |
| + TR-GRPO | **0.95** | **0.95** | **0.92** | **0.87** | **0.84** | **0.91** | ↑ 18.2% |

### 4.2.1 EXPERIMENTS ON K&K LOGIC PUZZLES

Table 1 reports results on the K&K Logic Puzzles benchmark, where TR-GRPO achieves significant improvements over GRPO across all difficulty levels. Notably, TR-GRPO boosts the average accuracy of Qwen2.5-3B from 39% (with GRPO) to 63%, representing a relative gain of over 60%. Similarly, for Qwen2.5-7B, TR-GRPO increases the average accuracy from 77% to 91% (+18%). It is worth noting that this task remains challenging even for powerful proprietary models such as GPT-4o, DeepSeek-R1, and o1, which achieve average accuracies of only 36–78%.

### 4.2.2 EXPERIMENTS ON MATH-RELATED DATASETS

To further demonstrate the effectiveness of our method, we conducted experiments on Math-related datasets. Table 2 summarizes performance on mathematical reasoning datasets. TR-GRPO consistently surpasses GRPO, with average gains of 6–10% across OlympiadBench, Minerva, MATH-500. AMC, and AIME. In particular, on AIME, TR-GRPO raises pass@16 from 16.5% to 18.1% and average@16 from 28.9% to 43.3%, demonstrating stronger robustness in symbolic and arithmetic reasoning. These results clearly highlight the superiority of our proposed TR-GRPO over GRPO baseline in advancing mathematical reasoning.

Table 2: Experimental results on math-related datasets (DSR for DeepScaleR and ORZ for Open Reasoner-Zero). The best results are indicated in **bold**.

| Dataset | Algorithms | Olympiad Bench | Minerva | MATH 500 | AMC avg@16 | AIME pass@16 | AIME avg@16 | Avg. | |
|---|---|---|---|---|---|---|---|---|---|
| **Qwen2.5-7B** | | 27.64 | 18.38 | 63.00 | 22.21 | 30.00 | 5.00 | 27.71 | |
| DSR | + GRPO | 36.50 | 29.66 | 74.67 | **47.72** | 28.89 | 16.46 | 38.98 | |
| | + TR-GRPO | **38.48** | **32.35** | **79.40** | 46.84 | **43.33** | **18.13** | **43.09** | ↑ 10.5% |
| ORZ | + GRPO | 38.23 | 27.69 | 78.33 | **49.57** | 32.22 | 12.92 | 39.83 | |
| | + TR-GRPO | **40.12** | **30.51** | **78.60** | 45.48 | **43.33** | **16.88** | **42.49** | ↑ 6.7% |

### 4.2.3 EXPERIMENTS ON AGENTIC VERLTOOL SEARCH

To extend our evaluation, we further examine TR-GRPO on agentic RLVR tasks. Table 3 presents results in the agentic RLVR setting, where TR-GRPO delivers the largest improvements. On knowledge-intensive QA tasks, TR-GRPO nearly doubles the performance of GRPO, improving average exact-match accuracy from 13.8% to 27.3% (+97%) for Qwen2.5-3B, and yielding a 14% relative gain on Qwen3-4B-Instruct-2507. These results confirm the effectiveness of token-level regulation in tool-augmented reasoning, where stability and precision are critical. Together, these results establish

TR-GRPO as a consistently stronger optimization method than GRPO across diverse RLVR domains, with particularly striking gains in logic puzzles and agentic reasoning.

Table 3: Experimental results for the agentic task using VT-Search on knowledge-QA benchmarks. † represents in-domain datasets and ⋆ represents out-domain datasets. The best results are indicated in **bold**.

| Model | General QA | | | Multi-hop QA | | | | | |
|---|---|---|---|---|---|---|---|---|---|
| | NQ | TriviaQA | PopQA | HQA | 2Wiki | Musique | Bamboogle | Avg. | |
| Qwen2.5-3B | 0.50 | 1.20 | 0.80 | 0.50 | 1.50 | 0.00 | 0.00 | 0.64 | |
| + GRPO | 13.50 | 29.40 | 11.50 | 14.50 | 16.30 | 1.30 | **10.40** | 13.84 | |
| + TR-GRPO | **41.80** | **52.30** | **39.10** | **24.50** | **20.30** | **5.00** | 8.00 | **27.29** | ↑ 97.2% |
| Qwen3-4B | 31.68 | 60.00 | 38.60 | 27.59 | 14.89 | 6.70 | 23.20 | 28.95 | |
| + GRPO | 46.48 | 59.84 | 40.39 | 35.54 | 28.87 | **10.26** | 27.20 | 35.21 | |
| + TR-GRPO | **48.23** | **64.11** | **46.00** | **36.60** | **30.89** | **10.26** | **41.60** | **40.18** | ↑ 14.1% |

## 4.3 ABLATION STUDIES

To better understand the role of token-level weighting in our framework, we conduct an ablation study on the K&K Logic Puzzles benchmark using Qwen2.5-3B-Instruct.

We compare TR-GRPO against several alternative weighting strategies, including: (i) **Equal weight**, which assigns every token a constant value of 1; (ii) **Random weight**, where each token weight is sampled uniformly from $[0.5, 1.5]$; and (iii) **Reverse weight**, which inverts the weights computed by our method. Specifically, if TR-GRPO assigns a weight $w$ to a token $t$ in sequence $y$, the reverse weighting assigns it $2 - w$. As shown in Table 4, TR-GRPO consistently achieves the best performance across all difficulty levels. Equal, Random, and Reverse weighting provide only modest gains, but are clearly outperformed by our probability-aware design.

Table 4: Ablation study for token weight estimation on Qwen2.5-3B-Instruct on the K&K Logic Puzzles benchmark. The best results are indicated in **bold**.

| Model | Difficulty by Num of People | | | | | Avg. |
|---|---|---|---|---|---|---|
| | 3 | 4 | 5 | 6 | 7 | |
| Qwen2.5-3B-Instruct | 0.09 | 0.10 | 0.03 | 0.05 | 0.02 | 0.06 |
| Equal weight | 0.64 | 0.47 | 0.35 | 0.30 | 0.21 | 0.39 |
| Random weight | 0.67 | 0.43 | 0.37 | 0.32 | 0.24 | 0.41 |
| Reverse weight | 0.59 | 0.46 | 0.35 | 0.33 | 0.20 | 0.39 |
| **TR-GRPO** | **0.76** | **0.76** | **0.61** | **0.59** | **0.44** | **0.63** |

## 5 CONCLUSION

In this work, we introduced TR-GRPO, a simple yet effective extension of GRPO that incorporates token-level weighting to enhance stability and effectiveness in reinforcement learning for reasoning. By assigning probability-aware token weights, TR-GRPO stabilizes gradient updates and ensures that critical high-probability tokens are emphasized. Our theoretical analysis established how token probabilities govern gradient norms, and our extensive experiments across logic, math, and agentic reasoning confirmed substantial and consistent improvements over the GRPO baseline. Together, these findings highlight the importance of regulating token-level contributions in reinforcement learning, and position TR-GRPO as a robust framework for enhancing the stability and reasoning capability of large language models.

**Limitations.** A primary limitation of TR-GRPO lies in the additional computational overhead introduced by token-level weighting. Specifically, each update requires estimating and applying weights for all generated tokens, which slightly increases the per-step training cost compared to standard GRPO. Nevertheless, as shown in Appendix C.2, the overhead remains within a practically acceptable range and does not hinder scalability to larger models or datasets.

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

# A   RELATED WORK

**Large-Scale Reasoning Models.**   Large language models (LLMs)  (Lambert et al., 2024; Gao et al., 2024; Team et al., 2025; Guo et al., 2025; Yang et al., 2025a) have recently achieved remarkable progress across a variety of natural language processing tasks. In particular, researchers are increasingly focused on enhancing their performance in reasoning-heavy domains such as mathematics (Cobbe et al., 2021; Hendrycks et al., 2024), coding (Jain et al., 2024), and scientific reasoning (Rein et al., 2024). rStar-Math (Guan et al., 2025) develops a self-evolved deep-thinking strategy that significantly boosts the reasoning capabilities of smaller LLMs. Snell et al. (Snell et al., 2024) introduce dense, process-based verifier reward models that adaptively refine a model's response distribution based on the test-time prompt to strengthen reasoning skills. OpenAI's O-series (Jaech et al., 2024) applies large-scale reinforcement learning to train models capable of solving complex reasoning tasks, achieving state-of-the-art results on multiple benchmarks.

**Reinforcement Learning for Large Language Model.**   Before the emergence of reasoning-focused systems such as OpenAI's O-series (Jaech et al., 2024), reinforcement learning (RL) had been widely applied in the form of reinforcement learning from human feedback (RLHF) to enhance large language models' (LLMs) instruction-following and preference alignment (Ouyang et al., 2022). RLHF approaches are typically divided into online and offline optimization. Online algorithms—including PPO (Schulman et al., 2017), GRPO (Shao et al., 2024), and REINFORCE (Williams, 1992)—update the model by generating outputs during training and receiving immediate reward signals. Offline variants such as DPO (Rafailov et al., 2023), SimPO (Meng et al., 2024), and KTO (Ethayarajh et al., 2024) optimize policies using pre-collected preferences from annotators or LLMs. While offline methods are generally more efficient, they often trail online training in terms of performance (Tang et al., 2024). More recently, *reinforcement learning with verifiable rewards* (RLVR) has been introduced as a promising paradigm for improving LLM reasoning, especially in domains like mathematics and programming. OpenAI o1 (Jaech et al., 2024) was the first to demonstrate that RL can successfully scale reasoning abilities. Subsequent models—including DeepSeek-R1 (Guo et al., 2025), Kimi-2 (Team et al., 2025), and Qwen3 (Yang et al., 2025a)—have either matched or surpassed its performance. Notably, DeepSeek-R1 highlights how strong reasoning can emerge from outcome-based optimization using online RL, particularly GRPO (Shao et al., 2024). These advances also inspired the "zero RL" paradigm, where reasoning capabilities are elicited directly from the base model without explicit RL fine-tuning. Building on this idea, follow-up work has proposed methods such as DAPO (Yu et al., 2025), VAPO (Yue et al., 2025), SimpleRLZoo (Zeng et al., 2025), and Open-Reasoner-Zero (Hu et al., 2025), which further explore RL-based reasoning at scale.

# B   IMPLEMENTATION DETAILS

## B.1   EXPERIMENTS SETUP

This appendix provides complete details of datasets, prompts, reward design, rollout/training configurations, and evaluation protocols for all three RLVR settings used in this paper: **Logic (K&K)**, **Math**, and **Agentic (VT-Search)**.

### B.1.1   LOGIC: KNIGHTS & KNAVES (K&K)

Following  Xie et al. (2025); Yang et al. (2025c), we adopt LLMs after instruction fine-tuning (`Qwen2.5-3B-Instruct` (Yang et al., 2025a) and `Qwen2.5-7B-Instruct-1M` (Yang et al., 2025b)) as the initialization point. The tailored prompt designed for the LLMs is provided below.

> **Prompt**
>
> system\n You are a helpful assistant. The assistant first thinks about the reasoning process in the mind and then provides the user with the answer. The reasoning process and answer are enclosed within <think></think> and <answer></answer> tags, respectively, i.e., <think> reasoning process here </think><answer> answer here </answer>. Now the user asks you to solve a logical reasoning problem. After thinking, when you finally reach a conclusion, clearly

> state the identity of each character within <answer></answer> tags. i.e., <answer> (1) Zoey is a knight\n (2) ... </answer>.\n user\n{problem}\n assistant\n<think>

To promote chain-of-thought (CoT) reasoning in LLMs, Xie et al. (2025) introduces a reward function with two main components, as shown in Table 5. The output is judged as fully correct if the LLMs generate CoT reasoning wrapped inside <think>...</think> tags, and the final prediction enclosed within <answer>...</answer> tags.

Table 5: Reward design for K&K (Logic-RL).

|  | Format Reward | Answer Reward |
|---|---|---|
| Completely Correct | 1 | 2 |
| Partially Correct | $-1$ | $-1.5$ |
| Completely Wrong | $-1$ | $-2$ |

### B.1.2 MATH-RELATED DATASET

As mentioned in Section 4.1, inspired by (Yang et al., 2025c), we carry out additional experiments on two math-focused datasets, DSR-Uniform ($10,000$ problems, evenly covering difficulty levels) and ORZ ($57.000$ problems). In line with prior studies, we adopt Qwen2.5-7B (Yang et al., 2025a) as the base model. In this setting, no instruction-tuned templates are applied; instead, we employ a simple prompt directly.

> **Prompt**
>
> {problem} Let's think step by step and output the final answer within \\boxed{}.

LLMs without post-training generally struggle to follow strict output formats. Consequently, format-related signals are not included during training. Moreover, math tasks usually admit only a single correct solution, making partial credit unnecessary. Hence, a binary reward scheme is sufficient: the model receives a reward of 1 for a correct answer and 0 otherwise.

### B.1.3 AGENTIC: VT-SEARCH (KNOWLEDGE-AUGMENTED QA)

Question answering tasks frequently demand access to external knowledge that goes beyond a model's parametric memory, especially for factual queries and multi-hop reasoning. To address this, we follow the **Search-R1** (Jin et al., 2025) configuration from **VerlTool** framework (Jiang et al., 2025) which incorporates a FAISS-based retrieval module, allowing agents to query a local knowledge base and extract the most relevant evidence for answering complex questions.

Building on prior work (Jin et al., 2025; Song et al., 2025), an E5 retriever (Wang et al., 2022) was employed with the 2018 Wikipedia dump (Karpukhin et al., 2020) as the indexed corpus. The agent alternates between retrieval operations and reasoning steps to form complete answers. we adopt Qwen2.5-3B (Yang et al., 2025a) and Qwen3-4B-Instruct-2507 (Yang et al., 2025a) as the base models.

For this task, we use accuracy as the main reward, defined as:

$$R_{\text{search}}(\mathbf{x}, \mathbf{y}) = \begin{cases} 1 & \text{if match}(\mathbf{y}, \mathbf{y}_g) \\ -1 & \text{otherwise} \end{cases} \tag{14}$$

For evaluation, we follow the VerlTool benchmark and report exact-match scores on both General Q&A benchmarks (NQ (Kwiatkowski et al., 2019), TriviaQA (Joshi et al., 2017), PopQA (Mallen et al., 2022)) and multi-hop Q&A benchmark (HotpotQA (Yang et al., 2018), 2Wiki (Ho et al., 2020), MuSiQue (Trivedi et al., 2022), Bamboogle (Press et al., 2022))

## B.2 HYPERPARAMETERS

The key hyperparameter configurations for GRPO training are summarized in Table 6. We adopt the *clip-higher* technique from DAPO (Yu et al., 2025) to stabilize entropy and prevent collapse. Checkpoints are saved every 20 RL steps, and full implementation details are available in our code release. For token importance estimation (Eq. 11 in the main paper), we set $\alpha = 2.0$ and $\mu = 0.25$ with clipping bounds $L = 1.0$ and $U = 1.4$. The scaling parameter is fixed at $\tau = 9.0$. All experiments are conducted on $8 \times$ NVIDIA H100 GPUs.

Table 6: Key hyperparameters for GRPO training.

| Hyperparameter | Logic (K&K) | Math | Agentic (Search–R1) |
|---|---|---|---|
| Group size per prompt $G$ | 8 | 8 | 8 |
| Sampling temperature | 0.7 | 1.0 | 0.8 |
| Max response length | 4096 | 4096 | 4096 |
| Optimizer / LR | AdamW / $1 \times 10^{-6}$ | AdamW / $1 \times 10^{-6}$ | AdamW / $1 \times 10^{-6}$ |
| KL penalty coefficient | 0.001 | 0.001 | 0.001 |
| PPO clip ratio (low / high) | 0.20 / 0.24 | 0.20 / 0.24 | 0.20 / 0.24 |
| Mini-batch size | 64 | 128 | 32 |
| Micro-batch size (updates) | 256 | 512 | 256 |

# C ADDITIONAL EXPERIMENT RESULTS

## C.1 HYPERPARAMETER SENSITIVITY ANALYSIS

In Eq. 11, the token weight is determined by five hyperparameters: $\alpha$, $\mu$, $L$, $U$, and $\tau$. Among these, $\alpha$ and $\mu$ mainly control the global mean of the weight distribution. Since the sigmoid term $\sigma(\text{sg}[\pi_\theta(o_{i,t} \mid q, o_{i,<t})]/\tau)$ lies in $(0.5, 1)$, we fix $\alpha = 2$ and $\mu = 0.25$, resulting in a centered range $(0.5, 1.5)$ with mean value close to $1.0$. This ensures that token weights remain stable around unity, while still allowing enough variation to emphasize or de-emphasize tokens depending on their probability.

We then validate the two remaining hyperparameters, $\tau$ and the clipping range $(L, U)$. Table 7 reports results on the K&K Logic Puzzles benchmark using Qwen2.5-3B-Instruct. For $\tau$, we sweep across $[0.5, 2.0, 7.0, 9.0, 10.0, 20.0]$. The results indicate that performance is quite sensitive to $\tau$, with very small or very large values leading to degradation. The best accuracy is achieved in a moderate range around $\tau = 9.0$, suggesting that excessive sharpening or flattening of the sigmoid output is suboptimal.

For $(L, U)$, we consider four ranges: $(1.0, 1.2)$, $(1.0, 1.4)$, $(0.8, 1.5)$, and $(0.5, 1.5)$. In contrast to $\tau$, the choice of clipping bounds has only a mild effect, and all settings achieve comparable performance. This suggests that the precise clipping range mainly acts as a safeguard against extreme values rather than a critical tuning factor.

## C.2 COMPUTATIONAL COSTS

Table 8 reports the computational overhead of applying TR-GRPO compared to the GRPO baseline on the K&K Logic Puzzles dataset. The only additional operation required by TR-GRPO is the computation of token-level weights. Importantly, these weights are derived directly from the model's own output probabilities and therefore do not require any auxiliary teacher models or extra forward passes. As a result, peak GPU memory usage remains essentially unchanged between GRPO and TR-GRPO across both the Qwen2.5-3B-Instruct and LLaMa3.1-8B-Instruct backbones.

In terms of runtime, TR-GRPO does introduce a modest increase in training time per sample (e.g., 271.4 vs. 286.0 minutes on Qwen2.5-3B-Instruct, and 678.5 vs. 722.2 minutes on LLaMa3.1-8B-Instruct). However, this overhead is relatively minor compared to the substantial performance improvements reported in Section 4. Together, these results confirm that TR-GRPO

Table 7: Hyperparameter sensitivity of token weight estimation on Qwen2.5-3B-Instruct on the K&K Logic Puzzles benchmark.

| Parameter | Value | 3 | 4 | 5 | 6 | 7 | Avg. |
|---|---|---|---|---|---|---|---|
| | 0.5 | 0.63 | 0.48 | 0.35 | 0.31 | 0.20 | 0.39 |
| | 2.0 | 0.64 | 0.49 | 0.37 | 0.33 | 0.19 | 0.40 |
| | 7.0 | 0.67 | 0.62 | 0.41 | 0.37 | 0.24 | 0.46 |
| $\tau$ | 9.0 | 0.76 | 0.76 | 0.61 | 0.59 | 0.44 | 0.63 |
| | 10.0 | 0.77 | 0.79 | 0.58 | 0.57 | 0.37 | 0.62 |
| | 20.0 | 0.74 | 0.68 | 0.51 | 0.46 | 0.35 | 0.55 |
| | (1.0, 1.2) | 0.77 | 0.76 | 0.60 | 0.58 | 0.40 | 0.62 |
| $(L, U)$ | (1.0, 1.4) | 0.76 | 0.76 | 0.61 | 0.59 | 0.44 | 0.63 |
| | (0.8, 1.5) | 0.74 | 0.75 | 0.57 | 0.62 | 0.42 | 0.62 |
| | (0.5, 1.5) | 0.70 | 0.65 | 0.59 | 0.60 | 0.44 | 0.60 |

Table 8: Computational cost comparison of GRPO and TR-GRPO on K&K Logic Puzzle Dataset.

| Procedure | Qwen2.5-3B-Instruct | | LLaMa3.1-8B-Instruct | |
|---|---|---|---|---|
| | **GRPO** | **TR-GRPO** | **GRPO** | **TR-GRPO** |
| Training Time/Sample (minutes) | 271.4 | 286.0 | 678.5 | 722.2 |
| Peak GPU Memory (GB) | 459.7 | 460.2 | 581.2 | 580.6 |

delivers consistent gains in reasoning performance without imposing significant additional computational costs.

### C.3 CROSS-MODEL VALIDATION

Beyond the Qwen family used in previous experiments—which has been the dominant backbone in recent reasoning research—we further validate TR-GRPO on other model families, including `Mistral` and `LLaMA`. In addition, we test on the base variant `Qwen2.5-3B-Base` instead of the usual instruction-tuned model. All experiments are conducted under the logic K&K setting, with results reported in Table 9.

As shown in Table 9, across all models, TR-GRPO consistently outperforms GRPO, demonstrating that our method generalizes beyond Qwen-Instruct models. This provides further evidence that the proposed token-weighted strategy is broadly applicable and not limited to a single model family.

## D  PROOF OF OUR THEORIES

**Derivation of Derivative of GRPO in Eq. (7)**

For $\hat{A}_{i,t} > 0$, the inner term of the sum relevant to $o_{i,t}$ reduces to

$$h\left(o_{i,t}\right) = \begin{cases} (1 + \epsilon_h)\,\hat{A}_{i,t} & \frac{\pi_\theta(o_{i,t})}{\pi_{\text{old}}(o_{i,t})} > 1 + \epsilon_h \\ \frac{\pi_\theta(o_{i,t})}{\pi_{\text{old}}(o_{i,t})}\hat{A}_{i,t} & \frac{\pi_\theta(o_{i,t})}{\pi_{\text{old}}(o_{i,t})} \le 1 + \epsilon_h \end{cases}$$

The derivative w.r.t. $\theta$ becomes

$$\nabla_\theta h\left(o_{i,t}\right) = \begin{cases} 0 & \frac{\pi_\theta(o_{i,t})}{\pi_{\text{old}}(o_{i,t})} > 1 + \epsilon_h \\ \frac{\pi_\theta(o_{i,t})}{\pi_{\text{old}}(o_{i,t})}\nabla_\theta \log \pi_\theta(o_{i,t})\hat{A}_{i,t} & \frac{\pi_\theta(o_{i,t})}{\pi_{\text{old}}(o_{i,t})} \le 1 + \epsilon_h \end{cases}$$

Table 9: Experimental results on the K&K Logic Puzzles benchmark on various models. The best results are indicated in **bold**.

| Model | Difficulty by Number of People | | | | | Avg. | |
|---|---|---|---|---|---|---|---|
| | **3** | **4** | **5** | **6** | **7** | | |
| Qwen2.5-3B-Instruct | 0.09 | 0.10 | 0.03 | 0.05 | 0.02 | 0.06 | |
| + GRPO | 0.64 | 0.47 | 0.35 | 0.30 | 0.21 | 0.39 | |
| + TR-GRPO | **0.76** | **0.76** | **0.61** | **0.59** | **0.44** | **0.63** | ↑ 61.5% |
| Qwen2.5-7B-Instruct-1M | 0.22 | 0.15 | 0.08 | 0.10 | 0.02 | 0.11 | |
| + GRPO | 0.91 | 0.91 | 0.77 | 0.65 | 0.61 | 0.77 | |
| + TR-GRPO | **0.95** | **0.95** | **0.92** | **0.87** | **0.84** | **0.91** | ↑ 18.2% |
| Qwen2.5-3B-Base | 0.14 | 0.04 | 0.02 | 0.01 | 0.02 | 0.05 | |
| + GRPO | 0.60 | 0.54 | 0.43 | 0.38 | **0.28** | 0.45 | |
| + TR-GRPO | **0.68** | **0.62** | **0.44** | **0.47** | 0.26 | **0.49** | ↑ 8.9% |
| Mistral-7B-Instruct-v0.3 | 0.05 | 0.01 | 0.00 | 0.02 | 0.00 | 0.02 | |
| + GRPO | 0.29 | 0.16 | 0.09 | 0.11 | 0.03 | 0.14 | |
| + TR-GRPO | **0.47** | **0.27** | **0.18** | **0.15** | **0.08** | **0.23** | ↑ 64.3% |
| LLaMa3.1-8B-Instruct | 0.08 | 0.00 | 0.00 | 0.00 | 0.00 | 0.02 | |
| + GRPO | **0.92** | 0.92 | 0.83 | 0.79 | 0.80 | 0.85 | |
| + TR-GRPO | 0.89 | **0.95** | **0.88** | **0.87** | **0.81** | **0.88** | ↑ 3.5% |

Combining with the KL term derivative, we gain

$$\nabla_\theta h\left(o_{i,t}\right) + \beta \pi_{ref}\left(o_{i,t}\right) \frac{\nabla_\theta \pi_\theta\left(o_{i,t}\right)}{\pi_\theta\left(o_{i,t}\right)^2} - \beta \nabla_\theta \log \pi_\theta(o_{i,t})$$

$$= \nabla_\theta h\left(o_{i,t}\right) + \beta \frac{\pi_{ref}\left(o_{i,t}\right)}{\pi_\theta\left(o_{i,t}\right)} \nabla_\theta \log \pi_\theta\left(o_{i,t}\right) - \beta \nabla_\theta \log \pi_\theta(o_{i,t})$$

For $\hat{A}_{i,t} < 0$, the inner term of the sum relevant to $o_{i,t}$ reduces to

$$h\left(o_{i,t}\right) = \begin{cases} (1-\epsilon_l)\,\hat{A}_{i,t} & \frac{\pi_\theta(o_{i,t})}{\pi_{old}(o_{i,t})} < 1 - \epsilon_l \\ \frac{\pi_\theta(o_{i,t})}{\pi_{old}(o_{i,t})}\,\hat{A}_{i,t} & \frac{\pi_\theta(o_{i,t})}{\pi_{old}(o_{i,t})} \geq 1 - \epsilon_l \end{cases}$$

The derivative w.r.t. $\theta$ becomes

$$\nabla_\theta h\left(o_{i,t}\right) = \begin{cases} 0 & \frac{\pi_\theta(o_{i,t})}{\pi_{old}(o_{i,t})} < 1 - \epsilon_l \\ \frac{\pi_\theta(o_{i,t})}{\pi_{old}(o_{i,t})} \nabla_\theta \log \pi_\theta(o_{i,t})\hat{A}_{i,t} & \frac{\pi_\theta(o_{i,t})}{\pi_{old}(o_{i,t})} \geq 1 - \epsilon_l \end{cases}$$

Combining with the KL term derivative, we gain

$$\nabla_\theta h\left(o_{i,t}\right) + \beta \pi_{ref}\left(o_{i,t}\right) \frac{\nabla_\theta \pi_\theta\left(o_{i,t}\right)}{\pi_\theta\left(o_{i,t}\right)^2} - \beta \nabla_\theta \log \pi_\theta(o_{i,t})$$

$$= \nabla_\theta h\left(o_{i,t}\right) + \beta \frac{\pi_{ref}\left(o_{i,t}\right)}{\pi_\theta\left(o_{i,t}\right)} \nabla_\theta \log \pi_\theta\left(o_{i,t}\right) - \beta \nabla_\theta \log \pi_\theta(o_{i,t})$$

Finally, leveraging the two above cases, we gain the final formula.

**Lemma D.1.** *Assume that $A_{1:m}$ is a sequence of matrices with $\sigma_{min}(A_i) \geq a_i^2 \geq 0$ and $\sigma_{max}(A_i) \leq b_i^2$ for all $i \in \{1, ..., m\}$. We then have*

$$\|x\|_2 \prod_{i=m}^{1} a_i \leq \left\| x \prod_{i=m}^{1} A_i \right\|_2 \leq \|x\|_2 \prod_{i=m}^{1} b_i$$

*Proof.* We start with
$$\|xA\|_2^2 = xAA^Tx^T.$$

According to the Rayleigh inequality, we have
$$\sigma_{\min}(A)\|x\|_2^2 \le \|xA\|_2^2 = xAA^Tx^T \le \sigma_{\max}(A)\|x\|_2^2$$

$$\sigma_{\min}(A)^{1/2}\|x\|_2 \le \|xA\|_2 \le \sigma_{\max}(A)^{1/2}\|x\|_2 \tag{15}$$

Consider $A = \prod_{i=m}^{1} A_i$ and apply Inequality 15 recursively, we obtain

$$\|x\|_2 \prod_{i=m}^{1} \sigma_{\min}(A_i)^{1/2} \le \|x \prod_{i=m}^{1} A_i\|_2 \le \|x\|_2 \prod_{i=m}^{1} \sigma_{\max}(A)^{1/2}$$

$$\|x\|_2 \prod_{i=m}^{1} a_i \le \|x \prod_{i=m}^{1} A_i\|_2 \le \|x\|_2 \prod_{i=m}^{1} b_i$$

$\square$

**Proof of Theorem 3.1**

We first have
$$\frac{\partial \log \pi_\theta(o_{i,t})}{\partial h_{i,t}} = 1_k - p_{i,t},$$

where $1_k$ is a one-hot vector over the vocabulary, the token $o_{i,t}$ has the index $k$ in the vocabulary, $p_{i,t}$ is the distribution over vocabulary with $\pi_\theta(o_{i,t}) = p_{i,t}(k)$.

Consider a specific layer $l$, we then have

$$\frac{\partial \log \pi_\theta(o_{i,t})}{\partial \theta_l} = \frac{\partial \log \pi_\theta(o_{i,t})}{\partial h_{i,t}} \frac{\partial h_{i,t}}{\partial a_L} \frac{\partial a_L}{\partial a_{L-1}} \cdots \frac{\partial a_{l+1}}{\partial a_l} \frac{\partial a_l}{\partial \theta_l}$$

$$= [1_k - p_{i,t}] W \prod_{i=l}^{L} J_i G_l.$$

Applying Lemma D.1, we reach

$$\|1_k - p_{i,t}\|_2 a^W a_l^G \prod_{i=l}^{L} a_i^J \le \|\frac{\partial \log \pi_\theta(o_{i,t})}{\partial \theta_l}\|_2 \le \|1_k - p_{i,t}\|_2 b^W b_l^G \prod_{i=l}^{L} b_i^J$$

We further bound

$$\|1_k - p_{i,t}\|_2 = \sqrt{(1 - p_{i,t}(k))^2 + \sum_{j \ne k} p_{i,t}(j)^2} \ge 1 - p_{i,t}(k) = 1 - \pi_\theta(o_{i,t}).$$

$$\|1_k - p_{i,t}\|_2 = \sqrt{(1 - p_{i,t}(k))^2 + \sum_{j \ne k} p_{i,t}(j)^2} \le \sqrt{(1 - p_{i,t}(k))^2 + \left(\sum_{j \ne k} p_{i,t}(j)\right)^2}$$

$$= \sqrt{(1 - p_{i,t}(k))^2 + (1 - p_{i,t}(k))^2} = \sqrt{2}(1 - p_{i,t}(k)) = \sqrt{2}(1 - \pi_\theta(o_{i,t})).$$

Therefore, we further reach

$$(1 - \pi_\theta(o_{i,t})) a^W a_l^G \prod_{i=l}^{L} a_i^J \le \|\frac{\partial \log \pi_\theta(o_{i,t})}{\partial \theta_l}\|_2 \le \sqrt{2}(1 - \pi_\theta(o_{i,t})) \|1_k - \pi_\theta(o_{i,t})\|_2 b^W b_l^G \prod_{i=l}^{L} b_i^J$$

Finally, by noting that

$$\|\nabla_\theta \log \pi_\theta (o_{i,t})\|_2^2 = \sum_{l=1}^{L} \|\frac{\partial \log \pi_\theta (o_{i,t})}{\partial \theta_l}\|_2^2,$$

hence leading to

$$\frac{1}{\sqrt{L}} \sum_{l=1}^{L} \|\frac{\partial \log \pi_\theta (o_{i,t})}{\partial \theta_l}\|_2 \leq \|\nabla_\theta \log \pi_\theta (o_{i,t})\|_2 \leq \sum_{l=1}^{L} \|\frac{\partial \log \pi_\theta (o_{i,t})}{\partial \theta_l}\|_2$$

we reach the conclusion.

**Proof of Theorem 3.2**

We have
$$\|g_{i,t}\|_2 = w_{it} |\gamma_{i,t}| \|\nabla_\theta \log \pi_\theta(o_{i,t})\|_2.$$
Using the bound for $\|\nabla_\theta \log \pi_\theta(o_{i,t})\|_2$ developed in Theorem 3.1, we reach the conclusion.

## E  USE OF LARGE LANGUAGE MODELS

In accordance with the ICLR 2026 policy, we disclose our use of Large Language Models (LLMs) during the preparation of this paper. LLMs were employed solely as writing assistants to aid in polishing the presentation, improving clarity, and shortening or rephrasing certain passages. Importantly, all scientific ideas, theoretical developments, model design, and experimental results reported in this paper are entirely conceived and executed by the authors. The LLM was never used to generate research concepts, hypotheses, or experimental findings. The authors take full responsibility for the content of the paper.

