# OpenReview forum: "Token-Regulated Group Relative Policy Optimization for Stable Reinforcement Learning in Large Language Models"
_ICLR.cc/2026/Conference — ICLR 2026 Conference Withdrawn Submission_

### Official Review · Reviewer_QQ5E · 2025-10-17

**Soundness:** 2
**Presentation:** 2
**Contribution:** 1
**Rating:** 2
**Confidence:** 5

**Summary:**

The authors claim to identify a critical issue—namely, that low-probability tokens disproportionately dominate gradient updates due to their inherently large gradient magnitudes—and propose a token-regulated method that downweights low-probability tokens while emphasizing high-probability ones.

However, neither the finding nor the proposed approach appears original. The manuscript’s theoretical analysis and methodology substantially overlap with prior work [1], yet this overlap is not properly acknowledged; the authors cite [1] only in the experimental setup, stating that they follow its settings.

[1] Yang et al. Do Not Let Low-Probability Tokens Over-Dominate in RL for LLMs. AI4MATH Workshop at ICML 2025.

**Strengths:**

1. The theoretical analysis presents some improvements over [1]. However, there are notational inconsistencies, and the analysis appears to be inspired by [1] without appropriate citation.

2. The experimental results—particularly on math-related datasets—outperform [1]. Nonetheless, [1] is not treated as a baseline.

**Weaknesses:**

1. Theorem 3.1 appears very close to Proposition 4.2 in [1], yet is not properly cited.

2. The proposed “Token-Regulated” method closely mirrors the “Advantage Reweighting” approach in [1], again without proper citation. A similar sigmoid-based reweighting method also appears in [1]’s public codebase.

3. A significant technical flaw concerns the regulation of the KL divergence term. After applying the proposed regulation, the KL divergence between π_θ and π_ref is nonzero even when the two distributions are identical. In addition, the expression for w_{i,t} is unnecessarily complex and lacks theoretical justification.

4. The notation in Theorems 3.1 and 3.2 is confusing; it appears that α should be a and β should be b.

**Questions:**

1. Please clarify the relationship between your work and [1]. Many components appear highly similar—including even the color schemes in figures—yet [1] is cited only in the experimental setup.

2. Why is the KL divergence term also regulated? What theoretical or empirical benefits does this provide? From a theoretical standpoint, this appears inappropriate.

3. Please provide direct experimental comparisons with [1]. Given the similarities, a proper head-to-head comparison is necessary.

**Details Of Ethics Concerns:**

The manuscript raises substantial concerns regarding originality and attribution. Key ideas, theoretical results, and methodological components appear to closely track [1] without adequate citation or clear differentiation.

Specifically:

1. Abstract overlap: The paper states that “low-probability tokens disproportionately dominate gradient updates due to their inherently large gradient magnitudes” (lines 016–020), which is nearly identical to [1]’s abstract: “low-probability tokens disproportionately influence model updates due to their large gradient magnitudes.” This constitutes the central finding of [1] and is presented here without clear attribution.


2. Theoretical overlap: Theorem 3.1 (lines 191–201) is highly similar to Proposition 4.2 in [1], with nearly identical notation. The proof sketch (lines 1011–1021) follows the proof outline of Proposition 4.2 in [1]. In addition, Eq. (7) matches Eq. (2) in [1] (including notation). These parallels are not properly cited, and the text appears to claim them as original contributions, which raises concerns of idea plagiarism.


3. Methodological overlap: The proposed “Token-Regulated” method (Eq. (9)) is almost the same as “Advantage Reweighting” in [1]. The primary differences are the specific weight form (Eq. (11)) and an additional term regulating the KL-divergence (Eq. (10)), which lacks theoretical justification. A similar weight form to Eq. (11) also appears in [1]’s public codebase.

4. Presentation overlap: Several figures and tables adopt formats that are nearly identical to those in [1].

Taken together, the evidence suggests the authors are well aware of [1], yet [1] is cited only to state that its experimental settings are followed. This level of acknowledgment is far below standard scholarly practice for attribution.

Conclusion: The manuscript appears to constitute idea plagiarism of [1], or potentially a dual submission of [1]. While the latter seems unlikely, the current version does not meet scholarly norms for originality and attribution.

---

### Official Review · Reviewer_LBTX · 2025-10-31

**Soundness:** 2
**Presentation:** 1
**Contribution:** 1
**Rating:** 0
**Confidence:** 4

**Summary:**

This paper proposes TR-GRPO, a probability-aware token reweighting scheme atop GRPO to dampen gradients from low-probability tokens and emphasize high-probability ones, reporting sizable gains on logic, math, and agentic RLVR tasks. However, its framing, theoretical motivation, and even experimental setup appear substantially overlapping with “Do Not Let Low-Probability Tokens Over-Dominate in RL for LLMs, 	arXiv:2505.12929” which already identifies the same dominance issue and introduces Advantage Reweighting and Lopti; the submission does not clearly position itself against that work.

**Strengths:**

1. Clear problem statement and intuitive fix; easy to implement in existing GRPO pipelines.
2. Consistent improvements across K&K logic, math suites, and agentic QA, with smoother training rewards.

**Weaknesses:**

1. Substantial overlap with prior work: same core phenomenon (low-probability tokens dominate updates), highly similar motivation and claims, and close experimental regimes (K&K, Minerva/MATH, AIME/AMC), yet no direct, head-to-head comparison.
2. Unclear novelty boundary: TR-GRPO's token weighting may be functionally similar to Advantage Reweighting and the two-stage Lopti schedule; equivalence or strict improvements are not established.

**Questions:**

1. Please explicitly compare with “Do Not Let Low-Probability Tokens Over-Dominate in RL for LLMs”: what is new beyond its phenomenon, proofs, and methods (Advantage Reweighting, Lopti)?  I also find similar experiment setup in supplementary materials.

**Details Of Ethics Concerns:**

found that this paper's background, motivation, proof, and methodology, as well as the code provided in the supplementary materials, are highly similar to *Do Not Let Low-Probability Tokens Over-Dominate in RL for LLMs*(arXiv:2505.12929). However, the authors did not detail in the paper the relationship and differences between this work and arXiv:2505.12929.

---

### Official Review · Reviewer_x5ev · 2025-11-01

**Soundness:** 1
**Presentation:** 2
**Contribution:** 1
**Rating:** 2
**Confidence:** 5

**Summary:**

This paper observes that in GRPO training, low-probability tokens tend to dominate gradient updates, causing instability. To address this, the authors propose a strategy-grouped update that partitions tokens by probability and updates them separately to balance learning signals. Experiments on math and logic puzzle tasks demonstrate more stable training and improved overall performance.

**Strengths:**

See weaknesses.

**Weaknesses:**

There appear to be concerns regarding the originality of this submission. The paper’s motivation, targeted problem, theoretical formulation, and even specific theorem statements are almost identical to those presented in “Do Not Let Low-Probability Tokens Over-Dominate in RL for LLMs” (arXiv:2505.12929). The resemblance extends beyond conceptual similarity: the theoretical derivations are nearly identical, and large portions of the codebase, including file structures and even the Conda environment name, appear to be copied from that work. Given the extent of textual, mathematical, and implementation overlap, this submission raises serious concerns about academic integrity and originality. I recommend that the program chairs carefully investigate this potential case of plagiarism before further consideration.

**Questions:**

See weaknesses.

**Details Of Ethics Concerns:**

There appear to be concerns regarding the originality of this submission. The paper’s motivation, targeted problem, theoretical formulation, and even specific theorem statements are almost identical to those presented in “Do Not Let Low-Probability Tokens Over-Dominate in RL for LLMs” (arXiv:2505.12929). The resemblance extends beyond conceptual similarity: the theoretical derivations are nearly identical, and large portions of the codebase, including file structures and even the Conda environment name, appear to be copied from that work. Given the extent of textual, mathematical, and implementation overlap, this submission raises serious concerns about academic integrity and originality. I recommend that the program chairs carefully investigate this potential case of plagiarism before further consideration.

---

### Official Review · Reviewer_S2cj · 2025-11-01

**Soundness:** 2
**Presentation:** 3
**Contribution:** 2
**Rating:** 2
**Confidence:** 3

**Summary:**

The paper is well motivated. However, the contribution of the proposed appraoch is not sound.

**Strengths:**

1. The paper's core motivation is exceptionally well-justified. The theoretical analysis (Theorem 3.1), which demonstrates that gradient norms scale with $1 - \pi_\theta(o_t)$, coupled with the empirical observation (Figure 1) that high-probability tokens are often critical logical or mathematical symbols, presents a clear and compelling case for intervention.

2. The experimental results across logic, math, and agentic tasks are extensive and demonstrate clear, consistent improvements over the GRPO baseline. The inclusion of a "Reverse" ablation study is particularly effective in validating the core hypothesis.

**Weaknesses:**

1. The proposed solution—the design of the weighting function—feels somewhat heuristic and lacks a strong theoretical derivation. While the choice of a sigmoid function and subsequent clipping is empirically effective, it appears as an engineered solution. The paper would be significantly strengthened by providing a more principled justification for this specific functional form, moving beyond the rationale that it is merely a smooth, increasing function of the token probability.

2. The paper's empirical analysis is primarily positioned against the established GRPO baseline and omits comparisons with other recent state-of-the-art RL methods for LLMs, such as DAPO. To fully establish TR-GRPO's contribution and competitiveness, it is crucial to include benchmarks against these other advanced optimizers. Such comparisons would provide a clearer understanding of its relative advantages and disadvantages within the current methodological landscape.

3. The proposed method modifies the objective by applying the token weight $w(\pi_\theta(o_t))$ directly to the importance ratio inside the $\min$ and $\text{clip}$ operations (Eq. 9). This approach intertwines the new weighting scheme with PPO's fundamental clipping mechanism, which was originally designed to constrain policy updates based on the unweighted importance ratio $r_t(\theta)$. The paper does not sufficiently analyze how this interaction affects the stability guarantees of the underlying PPO/GRPO algorithm. It is plausible that scaling the ratio by a weight $w > 1$ could push a larger fraction of tokens into the clipped region, potentially altering the optimization dynamics in unintended ways. A deeper discussion or analysis of this interaction is necessary to assure readers of the method's stability and theoretical soundness.

**Questions:**

How does the interaction (between the new weighting scheme and PPO's fundamental clipping mechanism) affects the stability guarantees of the underlying PPO/GRPO algorithm?

---

### Note · Authors · 2025-11-12

I have read and agree with the venue's withdrawal policy on behalf of myself and my co-authors.